# Telomere length and brain imaging phenotypes in UK Biobank

Anya Topiwala[1]*, Thomas E. Nichols[1,2], Logan Z. J. Williams[3], Emma C. Robinson[3], Fidel Alfaro-Almagro[4], Bernd Taschler[4], Chaoyue Wang[5], Christopher P. Nelson[6,7], Karla L. Miller[5], Veryan Codd[6,7], Nilesh J. Samani[6,7], Stephen M. Smith[5]

1 Nuffield Department Population Health, Big Data Institute, University of Oxford, Oxford, United Kingdom, 2 Nuffield Department of Clinical Neurosciences, Wellcome Centre for Integrative Neuroimaging, FMRIB, University of Oxford, Oxford, United Kingdom, 3 Centre for the Developing Brain, School of Biomedical Engineering and Imaging Sciences, King's College London, London, United Kingdom, 4 Nuffield Department of Clinical Neurosciences, Wellcome Centre for Integrative Neuroimaging (WIN FMRIB), University of Oxford, Oxford, United Kingdom, 5 Wellcome Centre for Integrative Neuroimaging (WIN FMRIB), Oxford University, Oxford, United Kingdom, 6 Department of Cardiovascular Sciences, University of Leicester, Leicester, United Kingdom, 7 NIHR Leicester Biomedical Research Centre, Glenfield Hospital, Leicester, United Kingdom

* anya.topiwala@bdi.ox.ac.uk

**Data Availability Statement:** Full pseudonymized participant data cannot be openly shared under the material transfer agreement with UK Biobank and ethics approval. Other researchers can apply for UK

## Abstract

Telomeres form protective caps at the ends of chromosomes, and their attrition is a marker of biological aging. Short telomeres are associated with an increased risk of neurological and psychiatric disorders including dementia. The mechanism underlying this risk is unclear, and may involve brain structure and function. However, the relationship between telomere length and neuroimaging markers is poorly characterized. Here we show that leucocyte telomere length (LTL) is associated with multi-modal MRI phenotypes in 31,661 UK Biobank participants. Longer LTL is associated with: i) larger global and subcortical grey matter volumes including the hippocampus, ii) lower T1-weighted grey-white tissue contrast in sensory cortices, iii) white-matter microstructure measures in corpus callosum and association fibres, iv) lower volume of white matter hyperintensities, and v) lower basal ganglia iron. Longer LTL was protective against certain related clinical manifestations, namely all-cause dementia (HR 0.93, 95% CI: 0.91–0.96), but not stroke or Parkinson's disease. LTL is associated with multiple MRI endophenotypes of neurodegenerative disease, suggesting a pathway by which longer LTL may confer protective against dementia.

## Introduction

Telomeres are protective caps at the end of chromosomes that progressively shorten with each cell division [1]. Their attrition is a marker of biological aging and may increase susceptibility to age-related diseases including Alzheimer's disease [2]. However, the mechanism by which accelerated cellular aging increases neuropsychiatric disease risk is unclear. Studies have shown structural changes in neurodegenerative diseases prior to detectable clinical symptoms [3]. MRI markers therefore provide intermediate endophenotypes to study the associations between biological aging and neuropsychiatric disease [4].

Biobank data to answer specific research questions. Further information about applying for data access can be obtained from the UK Biobank website (https://www.ukbiobank.ac.uk) or by emailing UK Biobank (ukbiobank@ukbiobank.ac. uk).

**Funding:** AT is supported by a Wellcome Trust (https://wellcome.org/) fellowship (216462/Z/19/ Z). CW is funded, in part, by the China Scholarship Council (CSC, https://www. chinesescholarshipcouncil.com/). SMS is supported by a Wellcome Trust Collaborative Award 215573/Z/19/Z. KLM is supported by a Wellcome Trust Senior Research Fellowship (202788/Z/16/Z). TEN is supported by the Li Ka Shing Centre for Health Information and Discovery, an NIH grant (https://www.nih.gov/, TN: R01EB026859), the National Institute for Health Research Oxford Biomedical Research Centre (BRC-1215-20014), and a Wellcome Trust award (TEN: 100309/Z/12/Z). The telomere length measurements were funded by the UK Medical Research Council (MRC), Biotechnology and Biological Sciences Research Council and British Heart Foundation through MRC grant MR/ M012816/1 to VC and NJS. VC and NJS are supported by the National Institute for Health Research (NIHR) Leicester Cardiovascular Biomedical Research Centre (BRC-1215-20010). FAA is funded by the UK Medical Research Council (MRC). LZJW is supported by the Commonwealth Scholarship Commission, United Kingdom. The funders had no role in study design, data collection and analysis, decision to publish, or preparation of the manuscript.

**Competing interests:** TN "Paid statistical consultancy, Perspectum". The other authors declare no competing financial interests. This does not alter our adherence to PLOS ONE policies on sharing data and materials.

Peripheral blood markers of cell aging are the most practical markers to obtain for epidemiological studies. To date, studies examining relationships between directly measured (as opposed to being imputed via single nucleotide polymorphisms) leucocyte telomere length (LTL) and neuroimaging markers [5] derived from limited number of brain regions; however, these have been implemented for small samples with limited power to detect subtle effects. The most consistent findings have been positive associations between LTL and total brain volume [5,6] and hippocampal volume [5,7,8]. These are of relevance to disease pathogenesis, as global and hippocampal atrophy are characteristic features of Alzheimer's disease [9]. However meta-analyses have not substantiated a hippocampal association [8,10]. A small trial of mental training (designed to cultivate presence, compassion and sociocognitive skills) found that longitudinal change in LTL was associated with changes in cortical thickness [11]. Furthermore, it remains unknown how LTL relates to other structural and functional brain measures, of relevance to neurological health [12]. These include grey matter structures, white matter microstructure, and functional connectivity. MRI endophenotypes are quantitative, and may lie closer to the underpinning genetic determinants than clinical phenotypes on the causal pathway to disease [13,14]. Determining relationships between telomere length and MRI markers could offer insights into biological mechanisms of neurodegenerative disorders.

Here we report the largest and most systematic investigation of LTL and brain structure and function to date. Associations of LTL were examined with 3,921 distinct univariate metrics from multi-modal MRI data as well as data-driven clusters of these image-derived phenotypes (IDPs), stringently controlling for confounders and multiple testing. Furthermore, we leveraged linked electronic health records to investigate LTL associations with relevant clinical phenotypes, dementia, stroke and Parkinson's disease.

## Methods

### Participants

Participants were drawn from UK Biobank [15], a prospective cohort study which recruited adults aged 40–69 years in 2006–10. We included subjects with usable brain imaging data released by 02.02.2021. All participants provided informed consent through electronic signature at baseline assessment.

The UK Biobank project was approved by the Northwest Haydock Research Ethics Committee (reference: 11/NW/0382).

### Scanning and MRI processing

Neuroimaging was performed at three centres (Newcastle upon Tyne, Stockport and Reading) on identical Siemens Skyra 3T scanners (software VD13) using a standard Siemens 32-channel head coil. We used all 6 MRI modalities: T1-weighted (T1w) and T2-weighted-FLAIR (T2w) structural imaging, susceptibility-weighted MRI (swMRI), diffusion MRI (dMRI), task functional MRI (tfMRI) and resting-state functional MRI (rfMRI) [12].

Full details of the pre-processing and quality control pipeline have been previously published [16]. In brief, T1w and T2w structural images were gradient distortion corrected and registered linearly and non-linearly (using FMRIB's Linear Registration Tool, FLIRT [17] and FMRIB's Nonlinear Image Registration Tool, FNIRT [18]) to standard MNI space. Brain extraction (using Brain Extraction Tool, BET [19], defacing and segmentation into tissue types (using FMRIB's Automated Segmentation Tool, FAST [20]) were then performed. Volumes for subcortical structures were generated by modelling using FMRIB's Integrated Registration and Segmentation Tool (FIRST [21]).

T1w and T2w volumes were also processed using FreeSurfer, generating cortical surface meshes and features [22]. White and pial surfaces were extracted with the FreeSurfer pipeline using–recon-all with–FLAIR and–FLAIRpial flags [23], due to improvements in pial surface placement compared to surface extraction using T1w images alone [24]. Anatomical regions of interest (ROIs) for each subject were generated as part of the FreeSurfer–recon-all pipeline through label propagation [25]. These ROIs were then used to summarise the following cortical surface features: cortical thickness, GWC and T1w/T2w ratio. GWC was calculated by FreeSurfer from T1w images using the formula: 100*(white-grey)/((white+grey)/2). To probe whether changes in intracortical myelin were driving GWC associations, we calculated new T1w/T2w ratio maps, which have been shown to correlate strongly with levels of intracortical myelin [26]. T1w/T2w maps were calculated using the volume-to-surface mapping algorithm available as part of [27] the Human Connectome Project (HCP) pipeline (https://github.com/Washington-University/HCPpipelines) [28]. Individual measures of average T1w/T2w ratio were calculated for ROIs and normalized to a reference ROI, in order to capture relative contrast in T1w/T2w ratio across the brain. The superior frontal cortex was chosen as the normalizing ROI, because of its relatively light myelination and minimal decline in myelination with age [29].

dMRI data were corrected for eddy currents, head motion and gradient distortion [16]. Using the tool DTIFIT (https://fsl.fmrib.ox.ac.uk/fsl/fslwiki/FDT) a diffusion tensor was fit (to the b = 1000 shell), generating fractional anisotropy (FA), tensor mode (MO), radial and mean diffusivities. FA images were fed into Tract-Based Spatial Statistics (TBSS [30]), which aligns the image onto a standard-space white matter skeleton. Additionally, dMRI was fed into NODDI (Neurite Orientation Dispersion and Density Imaging [31]) modelling to generate white matter microstructural parameters, including intra-cellular volume fraction (ICVF), isotropic water volume fraction (ISOVF) and orientation dispersion index (ODI). Skeletonised images were averaged within a set of standard-space tract masks to generate mean values.

The pipeline for rfMRI images used MELODIC (at dimensionalities 25 and 100) [32] which performs EPI unwarping, gradient distortion correction, head motion correction and high pass temporal filtering. Artefacts were removed using independent component analysis and FMRIB's ICA-based X-noiseifer (FIX [33]).

## Image-derived phenotypes

Here we mainly used summary IDPs generated by our team on behalf of UK Biobank and made available to researchers upon application. IDPs included: whole brain and cerebrospinal fluid (CSF) volumes, cortical volumes, surface area, thickness, and grey-white intensity contrast (GWC) from the T1w data for cortical regions, total volume of T2-FLAIR white matter hyperintensities extracted using BIANCA [34], $T2^*$ and quantitative susceptibility mapping (QSM) from swMRI data, white matter microstructural measures (including fractional anisotropy and diffusivity) from skeleton dMRI data, resting-state functional connectivity measures (standard deviations of network node timeseries–'amplitudes', as well as temporal correlations between nodes' timeseries–'edges') and task activation.

Two distinct and complementary metrics of brain iron deposition were used, $T2^*$ and magnetic susceptibility ($\chi$) [35], to produce IDPs. Whilst these metrics are somewhat distinct, consistent findings across the two provides greater evidence that iron levels are affected. Subject-specific masks for fourteen sub-cortical regions were derived from the T1w structural brain scan. We then calculated IDPs corresponding to the median $T2^*$ and magnetic susceptibility (from QSM) values for each region. $T2^*$ values were calculated from swMRI magnitude data. $T2^*$-induced signal decay was calculated from the two echo times. $T2^*$ images were spatially

filtered to reduce noise and linearly transformed into the T1w space. QSM depends on swMRI phase images, which were obtained from individual coil channels, combined, masked and unwrapped. χ was calculated using a QSM pipeline including background field removal, dipole inversion and CSF referencing [35]. Median χ values (in parts-per-billion) across voxels within each region were calculated.

## Telomere length measurements

LTL measurements were estimated from DNA collected at baseline using a well-validated qPCR assay. Measurements were reported as a ratio of the telomere repeat number to single-copy gene (T/S ratio). Multiple quality checks to control and adjust for technical factors were undertaken as described previously [36].

## Clinical measures

**Cognitive tests.** Cognitive testing was performed on all subjects at study baseline, and then subsets were invited to undertake additional testing at: 1) online follow up (mean 5.82 ±0.86 years after study baseline) and 2) imaging visit (**S1 Fig**). The cognitive battery differed at each of the three time points.

We selected the most clinically meaningful measures to examine associations with LTL. For trail-making tests (reflecting executive function: numerical–'TMT A'; alpha-numeric–'TMT B') duration was used. Tower rearranging measure was the number attempted (reflecting executive function). Digit span indicates the maximum digits recalled (reflecting working memory). Fluid intelligence score was a sum of correct answers (more meaningful than individual scores). Prospective memory was scored by number of incorrect answers. Pairs matching scored indicated the number correctly associated, reflecting visual memory. Matrix pattern completion was the duration spent answering each puzzle, reflecting processing speed. Reaction time was the mean time to correctly identify matches in a task based on the "Snap" card-game [37]. Digit substitution score indicated the number of correct symbols matched to numbers according to a key (assessing attention, visuospatial ability and associative learning).

**Neurodegenerative disease.** Neurodegenerative disease outcomes were algorithmically defined from UK Biobank's baseline assessment data collection (including self-report), linked data from Hospital Inpatient Statistics, primary care records and death records. The primary outcomes were incident dementia (all cause), stroke (all cause) and Parkinson's disease. Incident cases were defined as those with a diagnosis entered after study baseline, to reduce the risk of reverse causation. Prevalent cases (those already with a disease diagnosis at baseline) were excluded from clinical outcome analyses.

**Covariates.** For the imaging analyses, covariates were included in a two-step procedure. In analysis 1, adjustment was made for the full set of image-related confounds, including intra-cranial size, head motion, scanner table position, imaging centre, time between study baseline and imaging [38], in addition to age, age$^2$, age$^3$, sex and age*sex. In analysis 2, variables suggested to impact LTL and MRI measures in the literature were additionally added as potential confounds: smoking status [39] (reported as categorical variable: never/previous/current), body mass index [40] (calculated from measured height and weight), ethnicity (estimated using the top ten genetic ancestry components from principal component analysis), leucocyte count (derived from a blood sample at baseline), and weekly alcohol consumption [41] (in UK units estimated at baseline by summing across beverage types as previously described [42]).

For neurodegenerative disease and cognitive analyses, additional relevant confounds were added: educational qualifications (self-reported in categories), total household income, Townsend

Deprivation Index (continuous measure of deprivation based on census information), historical job type (coded according to the Standard Occupational Classification 2000 [43]).

## Statistical analyses

Analyses were performed in Matlab (R2019a) and R (version 3.6.0).

**Observational associations.** Initially, the full set of UKB-released 3,921 IDPs was used in an explorative analysis. Linear regression models were used to assess the relationship between LTL and individual IDPs at scan 1. Brain measures and LTL were quantile normalized resulting in Gaussian distributions with mean zero and standard deviation one. To adjust for multiple testing, both Bonferroni and (separately) false discovery rate (FDR) multiple comparison corrections were applied on 3921 tests.

Associations between LTL and latent factors of population covariation in IDPs, derived from independent component analysis (ICA) [44] were also examined. Such factors provide a lower-noise, more compact representation of the IDP data, and were derived following deconfounding, missing-data imputation and pre-whitening with principal components analysis (PCA, 1000 dimensions during imputation, with the top 20 then fed into ICA, https://fsl.fmrib.ox.ac.uk/fsl/fslwiki/FSLNets). ICA (n = 20 dimensions) was then run across IDP dimensions, using the FastICA algorithm [45]. IDP-weight-vectors are statistically independent of each other (by definition) and hence also orthogonal. Subject-weight-vectors are only restricted to being non-co-linear. All clusters are therefore distinct modes of population variation. Subject-weight vectors from the ICA components (or "clusters") were associated with LTL.

Cox proportional hazards models were used to estimate the association between LTL and dementia, stroke and Parkinson's disease incidence. The length of follow-up was calculated as interval between the baseline assessment and either date of event (first disease code for cases) or censoring (date of death if deceased during follow up or last date of data collection—2.1.22). The assumptions of proportional hazards were visually assessed using Schoenfeld residuals and formally tested by creating interactions with time for all predictors (S1 File). For covariates violating the proportional hazards assumption (age, BMI and historical job), stratified models were then fitted without the constraint of non-proportionality. Age and BMI were categorized into quintiles (strata) for these purposes. Separate baseline hazard functions were fitted for each strata. Restricted cubic splines (5 knots at $5^{th}$, $25^{th}$, $50^{th}$, $75^{th}$ and $95^{th}$ percentiles) were applied to LTL to assess for a non-linear relationship with dementia incidence. This model was compared to a linear model with a chi-squared test of log likelihoods between the two models. We also assessed the competing risk of death on the association between LTL and disease incidence using the subdistribution method [46].

## Results

31,661 participants had complete data and were included in the imaging analyses. The sample was 53.08% female with mean age 55.30±7.50 years old at baseline (**S1 Table**). Individuals were highly educated—almost half had a higher degree (49.46%) and experienced low levels of material deprivation (mean Townsend Deprivation Index (TDI) 1.92±2.71). Current smoking was rare (5.99%), but mean weekly alcohol intake was above national guidelines (mean 17.48 ±15.71 units). Very few amongst the imaged sample have developed dementia to date (0.17%). In comparison to the imaged sample, the wider UKB sample had higher material deprivation, lower levels of education, double the rates of smoking, and higher body mass index (BMI).

### Associations of LTL with brain phenotypes

LTL associations with MRI measures are summarized in **Table 1**.

**Table 1. Summary of individual image-derived phenotype associations with leucocyte telomere length.**

| Imaging modality | IDP type | Location | Association with LTL |
|---|---|---|---|
| T1 | Global grey matter and CSF volumes | Global | Positive |
| | Global white matter volumes | Global | Positive |
| | Regional volumes | Frontal, temporal, thalamus, brainstem, hippocampi, amygdala | Positive |
| | Cortical thickness | Middle and superior frontal sulcus, lingual sulcus, medial occipito-temporal sulcus; Superior temporal gyrus. | Positive |
| | Grey-White Contrast (GWC) | Lingual, pericalcarine, inferior and lateral parietal, precuneus, supramarginal | Negative |
| T1 and T2-FLAIR | White matter hyperintensity volume | Global and periventricular | Negative |
| swMRI | T2* | Caudate, putamen | Positive |
| | Susceptibility | | Negative |
| dMRI | FA, ICVF, MO | Corpus callosum splenium, sagittal stratum, inferior and superior longitudinal fasiculus, inferior fronto-occipital fasciculus | Negative |
| | L1-3, MD, ISOVF | | Positive |

Only associations significant after multiple testing correction are included. Abbreviations: FLAIR–fluid-attenuated inversion recovery, swMRI–susceptibility-weighted magnetic resonance imaging, DTI–diffusion tensor imaging, IDP–image-derived phenotype, CSF–cerebrospinal fluid, GWC–grey-white contrast, FA–fractional anisotropy, ICVF–intracellular volume fraction, MO–mode, MD–mean diffusivity, ISOVF–isotropic volume fraction.

**Individual IDP analyses.** LTL was associated with multiple IDPs across MRI modalities (**Fig 1** & **S2 Table**). Although associations were slightly attenuated, adjusting for additional (level 2) confounders did not make a material difference to the patterns of association. Longer LTL was associated with larger global (white (WM) and grey matter (GM)) and local brain volumes, including of frontal, temporal, thalamus, brainstem, hippocampi, and amygdalae regions. After adjusting for known confounders, LTL explained 0.05 (partial $R^2$) of the variance of peripheral grey matter volume. Longer LTL was associated with reduced grey-white contrast (GWC) in several regions, including lingual, pericalcarine, inferior and lateral parietal, precuneus and supramarginal regions (**Fig 2**). To elucidate whether higher GM signal or lower WM signal was driving the inverse relationship between LTL and GWC, we examined T1w/T2w ratio of the cortical ribbon [26]. There were no FDR-significant associations between LTL and T1w/T2w ratio, nor normalized T1w/T2w ratio by a reference ROI (**S3 Table**).

Associations with diffusion metrics were widespread. For the most part, longer LTL was associated with lower fractional anisotropy (FA), intracellular volume fraction (ICVF) and mode (MO), and higher L1-3, mean diffusivity (MD) and isotropic volume fraction (ISOVF). Such a pattern was observed in several tracts including the splenium of the corpus callosum, sagittal stratum, inferior and superior longitudinal fasiculus and inferior fronto-occipital fasciculus. In the fornix, the opposite pattern was noted with level 1 confounder adjustment–positive associations with FA, ICVF and MO, and negative associations with L1-3, MD and ISOVF. However associations with fornix DTI metrics became non-significant after level 2 confounders added to the models.

Negative associations between LTL and volume of white matter hyperintensities (WMH, as estimated by BIANCA [34]) were evident, and appeared driven by lower periventricular WMH. LTL was positively associated with T2* in the caudate, putamen and pallidum. In a post-hoc analysis, LTL was negatively associated with magnetic susceptibility (χ) in bilateral caudate and putamen but was not associated with pallidum χ (**S4 Table**). Associations with functional MRI IDPs were noticeably few.

**Clustered IDPs.** IDP clusters were identified by performing ICA on individual IDPs (**S5 Table**). In most cases, individual clusters reflected IDPs from a single imaging modality,

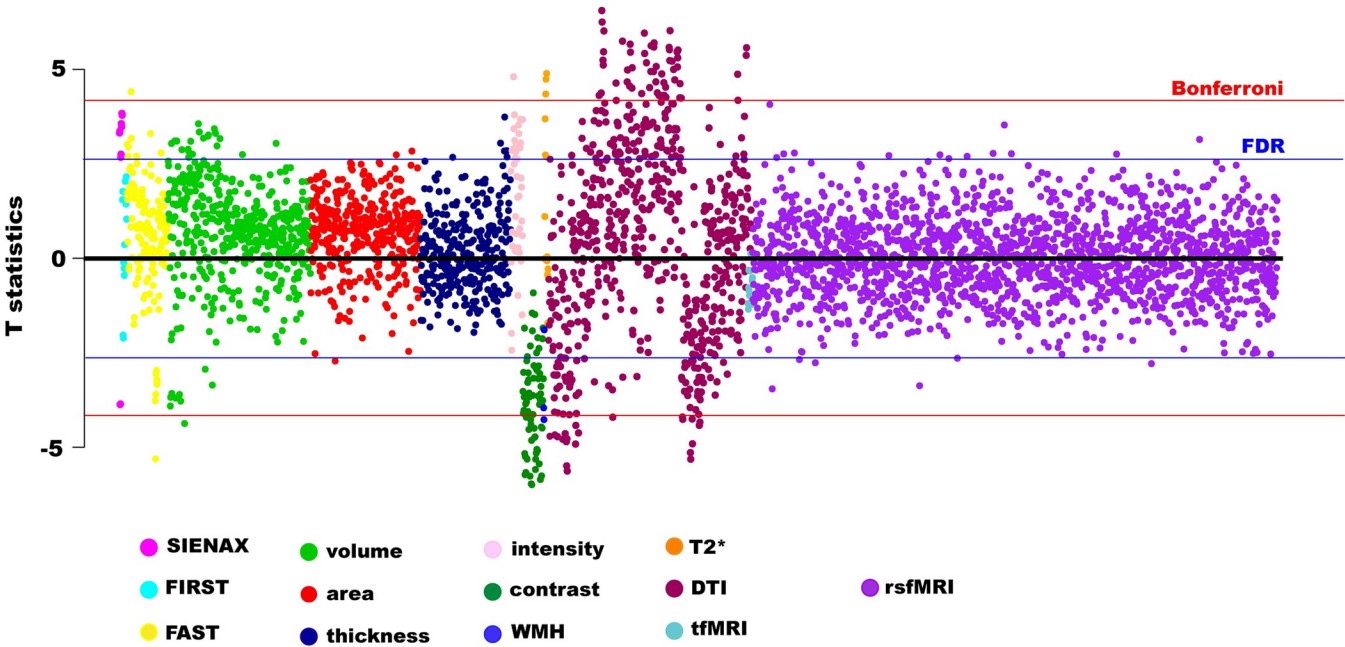

**Fig 1. Associations between leucocyte telomere length and individual imaging-derived phenotypes.** Estimates were generated from regression models adjusted for: full set of imaging-related confounders, age, age$^2$, age$^3$, sex, age*sex, body mass index, smoking, alcohol intake, leucocyte count, genetic ancestry. Blue line indicates False Discovery Rate threshold (3921 tests, p = 4.12x10$^{-3}$, T statistic = 2.64), red line indicates Bonferroni threshold (3921 tests, p = 1.28x10$^{-5}$, T statistic = 4.21). Abbreviations: IDP–image-derived phenotypes, dMRI–diffusion imaging, WMH–white matter hyperintensities, rsfMRI–resting state functional magnetic resonance imaging, tfMRI–task functional magnetic resonance imaging.

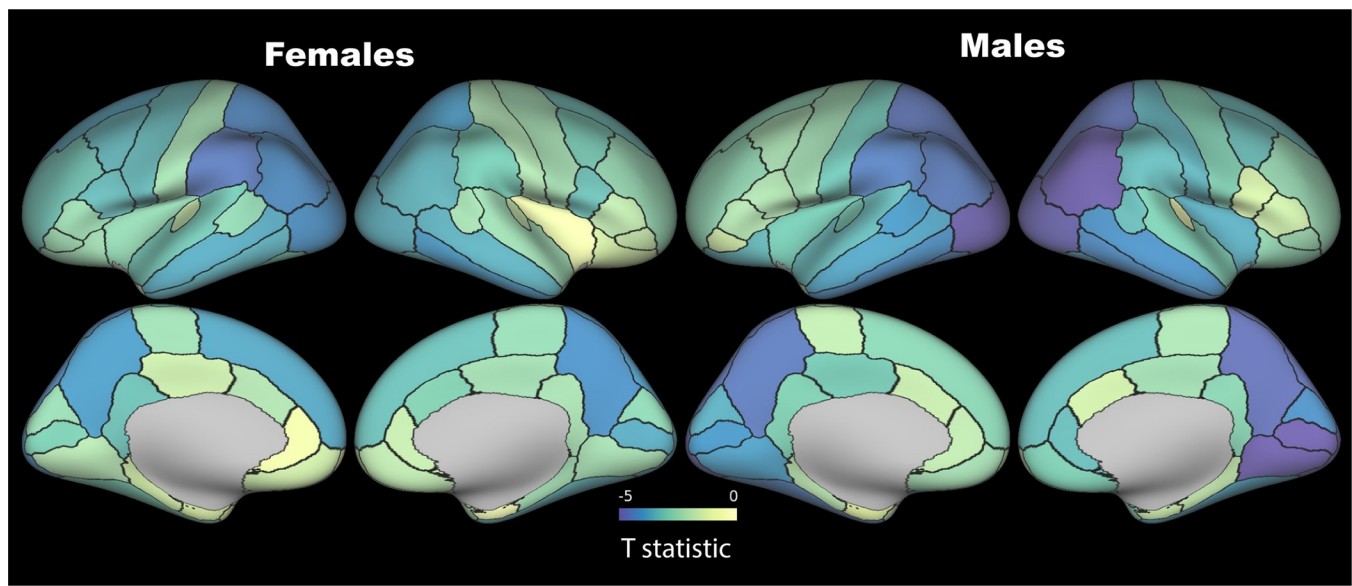

**Fig 2. Associations between grey-white matter contrast (on T1w) and leucocyte telomere length.** T statistics are shown projected onto the cortical surface as per the colour bar. False Discovery Rate threshold (3921 tests, T = 2.64), Bonferroni threshold (3921 tests, T = 4.21). Parcellations are based on the Desikan-Killiany atlas. Models adjusted for full set of imaging-related confounders, age, age$^2$, age$^3$, sex, age*sex, leucocyte count, smoking, alcohol intake, body mass index, genetic ancestry.

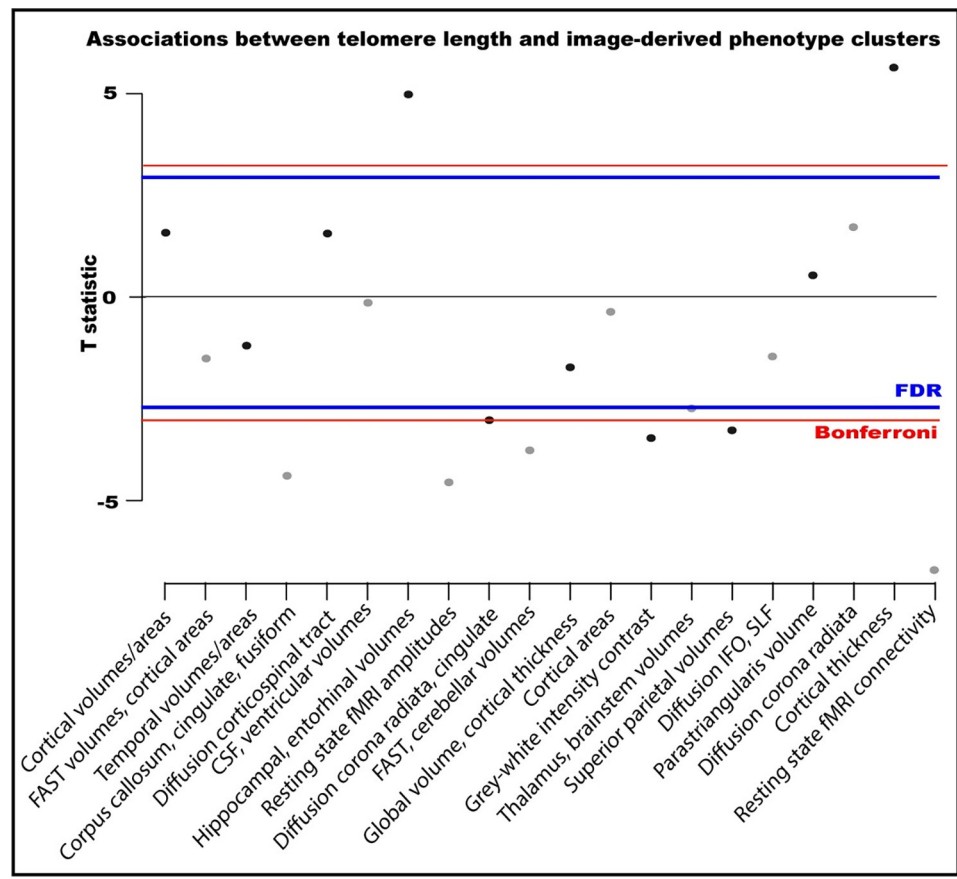

**Fig 3. Associations between leucocyte telomere length and imaging clusters from an independent components analysis-based dimensionality reduction on 3921 image-derived phenotypes.** Associations were adjusted for all imaging-related confounders, age, age², age³, sex, age*sex, smoking, alcohol intake, body mass index, leucocyte count, genetic ancestry. Constituents of clusters are in S5 Table. Multiple testing T statistic thresholds were calculated on the basis of 20 tests: False Discovery Rate (FDR) = 2.50, Bonferroni = 3.02.

with a few exceptions (clusters 6 & 10). LTL associated with ten of the twenty IDP clusters (**Fig 3**).

The strongest associations were with clusters comprising cortical thickness (cluster 19) and resting state fMRI connectivity (cluster 20) IDPs (**Fig 3**). Longer LTL associated with greater cortical thickness in the right hemisphere regions: globally, as well as in pre- and post-central, superior frontal, temporal and parietal areas. Cluster 20 is dominated by connectivity between network nodes ('edges') in motor, pre- and post-central and cerebellar networks (S5 Table). Longer LTL associated with higher functional connectivity between nodes in motor and sub-cortical-cerebellar networks (edges 101, 102, 103, 146 & 488), as well as between nodes within motor (edges 55 & 65) and subcortical-cerebellum networks (edges 151).

LTL was negatively associated with functional connectivity within the visual network (cluster 8), and corpus callosum (cluster 4), caudate (cluster 10) and superiorparietal area and volumes (cluster 15). Interestingly, LTL positively associated with hippocampal and entorhinal volumes (cluster 7) as well as fusiform areas (cluster 4).

## Clinical outcomes

During a median follow-up of 12.82 years (IQR: 12.08–13.54) for new onset disease, 5288 participants (1.42%) developed dementia, 5322 experienced a stroke (1.65%) and 1981 (0.61%)

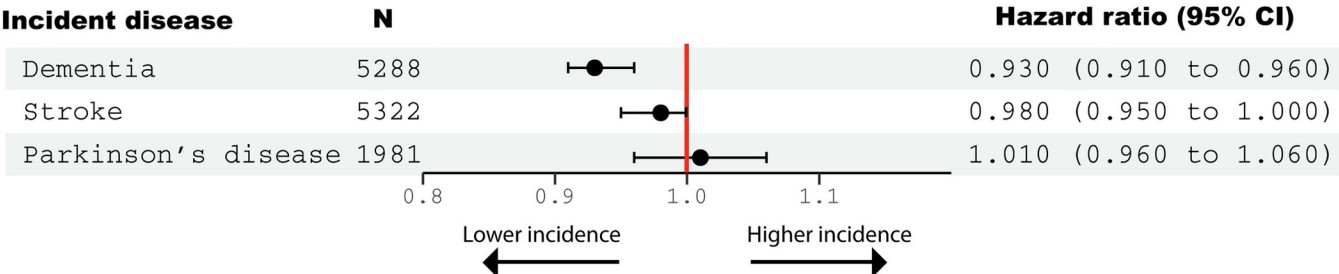

**Fig 4. Associations between leucocyte telomere length (quantile normalized) and incident disease during follow up.** Hazard ratios (95% confidence intervals) were generated from Cox proportional hazards models adjusted for: Age, age$^2$, age$^3$, sex, body mass index, educational qualifications, Townsend Deprivation Index, household income, historical job code, smoking, alcohol intake, leucocyte count, genetic ancestry.

developed Parkinson's disease within the larger UKB sample. There was overlap between clinical outcomes (12% of dementia cases had Parkinson's disease, and 11% had a previous stroke), as expected given Parkinson's and stroke are known pathways to dementia. Only 53 individuals within the imaging subsample developed dementia precluding exploration with imaging markers.

Longer LTL was significantly associated with reduced incidence of dementia during follow up (HR 0.93, 95% CI: 0.91–0.96) (Fig 4). Non-linear (restricted cubic splines) models offered no better fit than a linear model (p>0.05) (S2 Fig). Results were not much altered when accounted for the competing risk of death (HR = 0.96, 95% CI: 0.93–0.98).

Longer LTL was not significantly associated with a reduced incidence of stroke (HR 0.98, 95% CI: 0.95–1.00), nor with incident Parkinson's disease (HR 1.01 (0.96 to 1.06). There were no FDR-significant associations between LTL and cognitive test performance at any time point (S6 Table).

## Discussion

To our knowledge this is the largest and richest study to date examining relationships between LTL and MRI markers of brain structure and function.

Leucocyte telomere length was associated with multiple MRI phenotypes, including global and subcortical volumes, diffusion indices, GWC, WMH and markers of brain iron. Longer LTL was also associated with reduced incidence of dementia.

### Grey matter

LTL was positively associated with global, regional and subcortical grey matter volumes (including hippocampi and amygdalae). The findings are consistent with previous small studies which have examined a few regions of interest [6], and a recent meta-analysis of six studies [8]. The largest previous study in 1960 individuals observed positive associations with hippocampal, amygdala, and precuneus volumes [5]. Whilst some studies have similarly identified associations between the hippocampus and LTL attrition [7], others have not verified these findings [10]. We also observed positive associations of LTL with cortical thickness in multiple brain regions. Whilst most studies have not examined links with cortical thickness, one small longitudinal study did report short-term LTL lengthening with mental training associated with increases in cortical thickness [11].

We suggest three potential hypotheses for larger grey matter volumes in those with longer LTL [8]. First, those inheriting longer telomeres or losing less during growth may develop larger brains. Neurogenesis in humans (outside the hippocampus and subventricular zone) is

complete by gestational weeks 10–25, when telomerase is still prevalent [47,48]. Thus any LTL attrition from greater mitosis in embryos developing larger brains could be mitigated. Second, longer LTL may confer relative protection against brain aging, thus preserving volume. However our analyses were adjusted for age, reducing the likelihood that this explains our findings. Third, associations between grey matter volumes and LTL may result from genetic pleiotropy. A recent study found genetic colocalization between causal loci for LTL and global grey matter volumes [49]. Explicitly testing these hypotheses will require life course longitudinal studies.

### Diffusion measures

LTL was negatively associated with FA and positively with MD, L1-3, in the corpus callosum (splenium) and association fibres (sagittal stratum, inferior fronto-occipital fasciculus, superior and inferior longitudinal fasciculi). While much focus of telomere biology has been on LTL as a marker of accelerated aging, in fact LTL is mostly fixed by early adulthood. To give context, LTL shortens ~3000 base pairs during the first 20 years, and then just ~30 base pairs annually thereafter [50]. Consequently the majority of individuals maintain their LTL ranking during adulthood [51]. Early life processes are thus important to consider when interpreting our findings.

Interestingly, association tracts show the greatest FA increases in adolescence and early adulthood [52]. It is plausible that these FA increases result from greater myelination, which extends well into adolescence [53]. Myelination involves mitosis of oligodendrocyte precursor cells. TL shortens with each cell division and reflects a cell's replicative history [54]. Thus higher mitosis in tracts undergoing greater myelination could result in shorter TL compared to other brain regions. Telomerase is repressed after birth outside stem cells, so by adolescence could not counteract TL shortening in the affected tracts. This hypothesis is supported by observations that age-related TL decreases in white matter are greater than in grey matter [25,55]. Furthermore, TL shortening appears to be more marked in oligodendrocytes compared to astrocytes [56]. Here we measure leucocyte TL, which experiences one of the faster declines with age, due to the high proliferation rate of blood cells [57]. We propose that LTL is a closer representation to WM TL than GM TL.

### GWC contrast

In our study longer LTL was associated with reduced GWC, as assessed on T1 images. GWC is a ratio, and WM signal is higher than GM signal in T1w images; hence, either higher GM or lower WM signal could therefore drive decreases. Thinner cortex could blur the distinction between grey and white matter due to partial volume effects, which would reduce grey-white contrast. However, in our analyses longer LTL was associated with thicker cortex of at least some overlapping regions (e.g. lingual gyrus). As we found minimal association between LTL and T1w/T2w ratio in the cortical ribbon, we hypothesise that reduced WM signal, rather than intracortical myelination, is responsible. This fits with the inverse relationships between LTL and FA we observed in association fibres. Reductions in GWC have been observed in aging [58] and in some psychiatric disorders including psychosis and bipolar disorder [59,60]. In aging, signal intensity reduction in white matter is thought to be driving the reduced GWC associations, predominantly in lightly myelinated intracortical regions such as the frontal cortices [29]. Interestingly, we observed inverse associations between GWC and LTL in heavily myelinated sensory cortices. These included latero-occipital (object recognition), supramarginal (somatosensory association cortex), and pericalcarine (primary visual cortex). Such regions are myelinated earlier and are relatively protected from age-related degeneration [61]. This

supports our hypothesis that more heavily myelinated regions have undergone more mitotic divisions and thus experience greater TL shortening in early life.

## Other IDPs

Longer LTL was associated with reduced WMH volumes (WMH), particularly periventricular WMHs. Two previous studies have reported contradictory directions of association with LTL [6,62]. WMH increase in prevalence with aging, and are associated with increased risk of cerebrovascular disease and dementia [63,64].

Longer LTL associated with higher T2* and lower susceptibility in the putamen and caudate. Changes in myelin and iron have the same effect on T2*, but the opposite effects on susceptibility. Hence our findings suggest longer LTL associates with lower basal ganglia iron, which has relevance to neurodegenerative disease [65].

LTL was associated with clusters composed of resting state functional connectivity (within and between networks) more strongly than individual connectivity IDPs. We suspect either the clustering reduced noise allowing a signal to be detected, or alternatively, the effect is multivariate. We are aware of only one previous study of LTL and functional connectivity [44]. In 82 patients with mild cognitive impairment, widespread associations between LTL and connectivity were observed, particularly in frontal nodes and the cingulum. In contrast we observed negative associations between LTL and connectivity within the visual network, and positive associations with connectivity between motor and subcortical-cerebellar networks. Functional connectivity in motor networks are involved with voluntary movement [66], link with motor symptoms in Parkinson's disease [67], and correlate with motor recovery post-stroke [68]. Motor network connectivity is also affected by aging [69,70].

## Clinical outcomes

Longer LTL was associated with a reduced incidence of all-cause dementia. These findings are consistent with previous work showing shorter LTL as increasing Alzheimer's disease risk in observational [71,72] and Mendelian randomization studies [2,73]. Our non-significant association between LTL and incident stroke is consistent with findings from a large meta-analysis where solely prospective studies were examined [74]. To date, Mendelian randomization studies have not found evidence to support a causal relation either [75,76], although the statistical power may still be somewhat limited, given the relatively weak associations of genetic variants with phenotypes of interest.

LTL associated with larger global and subcortical (including hippocampal) GM volumes and increased cortical thickness in certain regions. Cortical thickness reductions predict cognitive decline [19]. Hippocampal atrophy is a biomarker of Alzheimer's disease [9]. WMH increase with aging and associate with cognitive deficits [77]. Brain iron accumulation has been linked to Alzheimer's disease [78] and Parkinson's disease [79]. We observed no associations between LTL and cognition that survived multiple testing correction, which is somewhat surprising given our observed protective effect on dementia incidence and MRI endophenotypes. The cognitive testing performed here was from a limited battery which may not have captured the relevant domains. Previous research examining LTL-cognition relationships has been conflicting. A meta-analysis of four prospective studies found no association between LTL and cognitive decline [80], although a more recent meta-analysis did suggest positive associations with attention and executive function [8]. A small study of patients with mild cognitive impairment did observe lower executive function in those with shorter LTL [44].

## Limitations

It is important to acknowledge several limitations. UK Biobank is subject to healthy volunteer bias, which may be further exacerbated within the imaging sample [81]. TL was measured in leucocytes, but the extent to which this reflects other organ tissues (such as tissues in the brain) is not clear [82]. Furthermore, causality is difficult to infer from cross-sectional analyses and our study did not directly assess telomere attrition. Multiple potential confounders were adjusted for but the possibility of residual confounding is a feature of all observational analyses. Dementia diagnosis were from electronic health records and as such do not have the granularity of a special study aimed at differentiating definite dementia subtypes.

## Conclusions

LTL associated with several MRI endophenotypes for neurodegenerative disease. These include: larger global and subcortical grey matter volumes, lower volume of white matter hyperintensities, lower basal ganglia iron deposition, and grey-white matter contrast of sensory cortices. Associations between LTL and brain structure provide a mechanism explaining the protective association of longer LTL on dementia incidence we observed. Accelerated cellular aging may represent a biological pathway to neurodegenerative disease.

## Supporting information

**S1 Fig. Summary of study timeline and analysis flowchart.** Abbreviations: LTL–leucocyte telomere length, MRI–magnetic resonance imaging, TMT–trail-making test.
(TIF)

**S2 Fig. Associations between leucocyte telomere length (LTL) and incident dementia.** Hazards are plotted relative to that of median telomere length. Restricted cubic splines (5 knots, quintiles) are applied to quantile normalized LTL. Cox proportional hazards models adjusted for: age, $age^2$, $age^3$, sex, body mass index, educational qualifications, Townsend Deprivation Index, household income, historical job code, smoking, alcohol intake, leucocyte count, genetic ancestry.
(TIF)

**S1 Table. Baseline characteristics of samples.**
(XLSX)

**S2 Table. Associations between leucocyte telomere length and individual image-derived phenotypes.**
(XLSX)

**S3 Table. Associations between leucocyte telomere length and T1/T2 contrast means and medians.**
(XLSX)

**S4 Table. Associations between leucocyte telomere length and quantitative susceptibility mapping image-derived phenotypes.**
(CSV)

**S5 Table. Image-derived phenotype cluster weights and associations with leucocyte telomere length.**
(XLSX)

**S6 Table. Associations between leucocyte telomere length and cognitive test performance.** (XLSX)

**S1 File. Testing proportional hazards assumptions.** (DOCX)

## Author Contributions

**Conceptualization:** Anya Topiwala, Thomas E. Nichols, Veryan Codd, Nilesh J. Samani, Stephen M. Smith.

**Data curation:** Stephen M. Smith.

**Formal analysis:** Anya Topiwala, Thomas E. Nichols, Logan Z. J. Williams, Fidel Alfaro-Almagro, Chaoyue Wang, Christopher P. Nelson, Karla L. Miller.

**Funding acquisition:** Stephen M. Smith.

**Investigation:** Anya Topiwala, Veryan Codd, Stephen M. Smith.

**Methodology:** Anya Topiwala, Thomas E. Nichols, Logan Z. J. Williams, Emma C. Robinson, Fidel Alfaro-Almagro, Bernd Taschler, Chaoyue Wang, Christopher P. Nelson, Karla L. Miller, Veryan Codd, Nilesh J. Samani, Stephen M. Smith.

**Supervision:** Nilesh J. Samani, Stephen M. Smith.

**Visualization:** Anya Topiwala.

**Writing – original draft:** Anya Topiwala.

**Writing – review & editing:** Anya Topiwala, Thomas E. Nichols, Logan Z. J. Williams, Emma C. Robinson, Fidel Alfaro-Almagro, Bernd Taschler, Chaoyue Wang, Karla L. Miller, Veryan Codd, Nilesh J. Samani, Stephen M. Smith.

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
