## [Decision Letter · Decision Letter 0]

23 Jan 2023

PONE-D-22-27282Telomere length and brain imaging phenotypes in UK BiobankPLOS ONE

Dear Dr. Topiwala,

Thank you for submitting your manuscript to PLOS ONE. After careful consideration, we feel that it has merit but does not fully meet PLOS ONE’s publication criteria as it currently stands. Therefore, we invite you to submit a revised version of the manuscript that addresses the points raised during the review process.

We look forward to receiving your revised manuscript.

Kind regards,

Pew-Thian Yap

Academic Editor

PLOS ONE

Journal Requirements:

2. Please ensure that you have specified (1) whether consent was informed and (2) what type you obtained (for instance, written or verbal, and if verbal, how it was documented and witnessed). If your study included minors, state whether you obtained consent from parents or guardians. If the need for consent was waived by the ethics committee, please include this information.

Funding: AT is supported by a Wellcome Trust (https://wellcome.org/) fellowship (216462/Z/19/Z). CW is funded, in part, by the China Scholarship Council (CSC, https://www.chinesescholarshipcouncil.com/). SS is supported by a Wellcome Trust Collaborative Award 215573/Z/19/Z. KLM is supported by a Wellcome Trust Senior Research Fellowship (202788/Z/16/Z). TN is supported by the Li Ka Shing Centre for Health Information and Discovery, an NIH grant (https://www.nih.gov/, TN: R01EB026859), the National Institute for Health Research Oxford Biomedical Research Centre (BRC-1215-20014), and a Wellcome Trust award (TN: 100309/Z/12/Z). The telomere length measurements were funded by the UK Medical Research Council (MRC), Biotechnology and Biological Sciences Research Council and British Heart Foundation through MRC grant MR/M012816/1 to VC and NJS. VC and NJS are supported by the National Institute for Health Research (NIHR) Leicester Cardiovascular Biomedical Research Centre (BRC-1215-20010). The funders had no role in study design, data collection and analysis, decision to publish, or preparation of the manuscript. 

However, funding information should not appear in the Acknowledgments section or other areas of your manuscript. We will only publish funding information present in the Funding Statement section of the online submission form. 

AT is supported by a Wellcome Trust (https://wellcome.org/) fellowship (216462/Z/19/Z). CW is funded, in part, by the China Scholarship Council (CSC, https://www.chinesescholarshipcouncil.com/). SS is supported by a Wellcome Trust Collaborative Award 215573/Z/19/Z. KLM is supported by a Wellcome Trust Senior Research Fellowship (202788/Z/16/Z). TN is supported by the Li Ka Shing Centre for Health Information and Discovery, an NIH grant (https://www.nih.gov/, TN: R01EB026859), the National Institute for Health Research Oxford Biomedical Research Centre (BRC-1215-20014), and a Wellcome Trust award (TN: 100309/Z/12/Z). The telomere length measurements were funded by the UK Medical Research Council (MRC), Biotechnology and Biological Sciences Research Council and British Heart Foundation through MRC grant MR/M012816/1 to VC and NJS. VC and NJS are supported by the National Institute for Health Research (NIHR) Leicester Cardiovascular Biomedical Research Centre (BRC-1215-20010). The funders had no role in study design, data collection and analysis, decision to publish, or preparation of the manuscript.

TN “Paid statistical consultancy, Perspectum”. The other authors declare no competing financial interests. 

Reviewers' comments:

Reviewer's Responses to Questions

**Comments to the Author**

1. Is the manuscript technically sound, and do the data support the conclusions?

Reviewer #1: Yes

Reviewer #2: Yes

2. Has the statistical analysis been performed appropriately and rigorously? 

Reviewer #1: Yes

Reviewer #2: Yes

3. Have the authors made all data underlying the findings in their manuscript fully available?

Reviewer #1: No

Reviewer #2: No

4. Is the manuscript presented in an intelligible fashion and written in standard English?

Reviewer #1: Yes

Reviewer #2: Yes

5. Review Comments to the Author

Reviewer #1: In this work association analyses between leucocyte telomere length (LTL) and brain multi-modal MRI traits were carried out based on the 31,661 UK Biobank participants. As the largest and richest study to date examining relationships between LTL and MRI markers of brain structure and function. this work includes many interesting conclusions and findings. I only have the following questions.

1) For the statistical method part, the proportional hazards models were carried out; and "assumptions of

proportional hazards were visually assessed using Schoenfeld residuals and

formally tested by creating interactions with time for all predictors. For

covariates violating the proportional hazards assumption (age, BMI and

historical job), stratified models were then fitted without the constraint of nonproportionality.

"

However, I cannot find details of the results based on the description of the steps in the proportional hazards models. For example, what is the results to check assumptions of proportional hazards or what did the Schoenfeld residuals look like; for which analyses the stratified models were performed. More details were needed to be added as supplementary text.

2) In all association analyses, age, sex, age^2, age^3 among many other confounding covariates were considered. However, age*sex was not considered. From previous literature, age*sex can be an important confounding factor for brain structural and functional metrics. If age*sex is included how much change will be made for the findings?

3) In Figure 2, is there a significance threshold for the vertex-wise t statistics map? We need to know the significance region.

4) Repeated colors were used to represent different categories in Figures 1 and S2. For example, in Figure 1, both the "FIRST" and "rfsMRI" were shown in purple, and "area" and "dMRI" were shown in red. Please change.

Reviewer #2: Topiwala et al. presents a comprehensive analysis to evaluate the association between leucocyte telomere length and multi-modal brain imaging derived phenotypes in 31,661 UK biobank participants. One of the striking finding is that the authors show that LTL has protective effects against dementia. I have following comments.

1. Figure 1: it would be helpful to use triangles to indicate the effect directions (positive or negative).

2. It’s hard to follow the paragraph of “Clustered IDPs”. Are these IDP clusters highly correlated with image derived phenotypes, or orthogonal complement to each other? Is each cluster representative of a specific imaging phenotype? How

to interpret the edges in networks?

3. In the follow-up study, the authors should exclude other diseases from controls.

4. What’s the associations between neurological and psychiatric disorder phenotypes? (They share strong genetic correlations.)

6. PLOS authors have the option to publish the peer review history of their article (what does this mean?). If published, this will include your full peer review and any attached files.

Reviewer #1: No

Reviewer #2: No

---

## [Author Response · Author response to Decision Letter 0]

8 Feb 2023

Please attached reviewer rebuttal

---

## [Editor Report · Decision Letter 1]

14 Feb 2023

Telomere length and brain imaging phenotypes in UK Biobank

PONE-D-22-27282R1

Dear Dr. Topiwala,

We’re pleased to inform you that your manuscript has been judged scientifically suitable for publication and will be formally accepted for publication once it meets all outstanding technical requirements.

Kind regards,

Pew-Thian Yap

Academic Editor

PLOS ONE
---

## [Editor Report · Acceptance letter]

24 Feb 2023

PONE-D-22-27282R1 

Telomere length and brain imaging phenotypes in UK Biobank 

Dear Dr. Topiwala:

I'm pleased to inform you that your manuscript has been deemed suitable for publication in PLOS ONE. Congratulations! Your manuscript is now with our production department. 

Kind regards, 

on behalf of

Dr. Pew-Thian Yap 

Academic Editor

PLOS ONE